

# A classification system for identifying persons with an unknown cardiovascular disease (CVD) status for a multiracial/ethnic Caribbean population

Amalia Hosein[1,2], Valerie Stoute[3] and Natasha Singh[4]

[1] Centre for Maritime and Ocean Studies, University of Trinidad and Tobago, Chaguaramas, Trinidad and Tobago
[2] Biomedical Engineering Programme, The University of Trinidad and Tobago, Point Lisas, Couva, Trinidad and Tobago
[3] V.A.S. Consulting, Port-of-Spain, Trinidad and Tobago
[4] Department of Bioscience, Thompson Rivers University, Kamloops, Canada

Corresponding author
Amalia Hosein,
amalia.hosein@utt.edu.tt

## ABSTRACT

**Background.** The need for classification systems for cardiovascular disease (CVD) that is population-specific is important towards understanding the clinical disease and diagnostics associated with the disease. This paper presents the form and validation results of this classification system.

**Method.** The survey data used was captured from 778 participants, 526 persons with no prior CVD, and 252 who reported prior CVD. Binomial logistic regression and Discriminant analysis were utilised to develop classification models. This classification system provided a general measure of severity of disease by utilising scores estimated from two algorithms developed from 13 routine physiologic measurements, along with demographic information of age and ethnicity, *inter alia*, and previous health status.

**Results.** For each model, specific score ranges were identified, which gave the best classification for those with a prior CVD incident (higher scores) and for others labelled as non-CVD (lower scores). The two classification models (Logistic Regression Model and Discriminant Analysis Model) developed had high area under the receiver-operating characteristic (AUROC) values (98% & 99%) and sensitivity (86 and 90%), which improved discrimination between Non-CVD and CVD participants and, more importantly, correctly classified a greater proportion of CVD participants. New to this type of research was the estimation and detailed evaluation of a range of scores, labelled non-differentiating, which fell in the middle of the spectrum and which contained the higher-end scores for the non-CVD individuals and the lower-end scores for CVD patients, all of whom were incorrectly classified, based on their prior history.

**Conclusion.** The classification system of scores is able to differentiate the CVD status of individuals, with good predictability, and could assist physicians with recommending different treatment plans. The two models in this classification system each individually outperformed the three established models in terms of the strength of their correct classifications of individuals with or without prior reported CVD incidents. More importantly, they have smaller non-differentiating ranges than the three known models and, in that range, the two new models have lower CVD/non-CVD ratios suggesting they are more likely to misclassify non-CVD individuals compared to CVD patients, which is a more benign misclassification. Further, when used in

combination, the two models increased the sensitivity, in classifying individuals of different ethnicities, beyond that of either one used independently or of any of the three standard European/North American models. These efforts will be instrumental in advancing personalised CVD management strategies and improving health outcomes across diverse populations.

## INTRODUCTION

In preventative management of cardiovascular diseases (CVD), risk prediction models such as the Framingham, ASSIGN, QRISK2 *etc.* have been developed and validated in North American and European populations (*Damen et al., 2016*). These epidemiologic risk models, however, may not always account for variations which exist among regions and countries due to different lifestyles, socio-economic conditions, and genetic predisposition (*Liew et al., 2018*; *Siontis et al., 2012*; *Sisa, 2018*; *Teo & Rafiq, 2021*). It has also been recognised that risk models may perform differently in populations of different and multiple racial or ethnic backgrounds (*Anand, Bradshaw & Prabhakaran, 2020*; *D'Agostino et al., 2001*; *Liu et al., 2004*). Thus, systematic efforts for model validation in other populations are essential. An evaluation of the ASSIGN, Framingham and QRISK®2 risk models in the Caribbean territory of Trinidad and Tobago, showed that these models should be utilized with caution since CVD patients are two (ASSIGN), 2.6 (QRISK2) and 3.7 (Framingham) times more likely than a non-CVD person to be misclassified (*Hosein et al., 2020*). Many clinicians, however, still utilise the risk models to guide their diagnostics (*Sisa, 2018*; *Sofogianni et al., 2022*).

A population-based approach to CVD risk scores is advantageous since existing risk scores seem to perform poorly in the developing countries and may lead to misclassification of individuals who do or do not require treatments (*Betts et al., 2019*; *Chamnan & Aekplakorn, 2017*). Many developing countries only describe estimated cardiovascular risk by calculating scores obtained from applying one or the other existing CVD risk estimation models to their population's cross-sectional data. Several countries have validated and recalibrated existing risk scores but only a few have developed new risk models and score ranges specific to their populations (*Assmann, Cullen & Schulte, 1998*; *Kasim et al., 2023*; *Wu et al., 2006*; *Yamwong, 2005*).

The Caribbean population is made up of a diaspora of cultures with an ethnic population distribution of 73% with African heritage, 6% with East Indian heritage, 8% that are multiracial, 1% of European/Indigenous peoples/Chinese origin and 11% of other ethnicities. In the Caribbean community, CVD is the largest contributor to mortality and morbidity accounting for 32% of all deaths (26–36%) per year (*WHO, 2016b*). The decision about whether to initiate specific preventive action and to what extent is mainly guided by CVD risk prediction models (*Payne, 2012*).

In Trinidad and Tobago, which has possibly the most pronounced multi-ethnic population in the Caribbean, with almost equal numbers of Afro-, Indo- and Mixed-Trinbagonians, mortality is due mainly to non-communicable diseases (primarily heart disease, diabetes, hypertension and cancer). The age-adjusted mortality rate from non-communicable diseases was 437.1 per 100,000 population (*PANO, 2021*). In 2019, the percentage distribution of non-communicable diseases reported was 82.7%, an increase from 79.3% in 2019. This suggests that health expenditure may need to concentrate more on awareness and preventative measures for these diseases (*GoTT, 2015*; *PANO, 2021*).

This paper presents the development of proprietary CVD classification models for a multi-ethnic/racial Trinidad and Tobago sample using three different approaches: chi-square automatic interaction detection (CHAID), logistic regression, and discriminant analysis. These methods were used, in the first instance, as classifiers and not future predictive models. They are cross-sectional designs which use Chi-square test branching (CHAID), a regression equation (logistic regression) and a discriminant group algorithm (discriminant analysis) to estimate significant classifying variables. CHAID analysis can identify significant variables but does not specifically quantify their relative impact, which hinders full and independent quantitative model development *via* this approach. On the other hand, both logistic regression and discriminant analyses were used to extract algorithms which yielded classification scores for each member of the sample. All three approaches gave classification percentages of correctly classified and misclassified members for each of the CVD and non-CVD groups. Only the two scoring models, however, were used to make detailed score classification performance comparisons with the three established CVD risk prediction models, ASSIGN, Framingham and QRISK2, in discriminating between CVD and non-CVD participants.

## MATERIALS & METHODS

### Study sample - TT2015 dataset: 10.6084/m9.figshare.21551550)

Data was collected as previously described in *Hosein et al. (2020)* where survey and biometric information was collected from 778 participants between the ages of 18 and 75 during the period October 2013 to May 2014. The study sample included 526 participants from the general population, classed as non-CVD participants; and 252 patients from four different hospitals with a diagnosed CVD, who were classed as CVD participants. CVD patients were defined as those with any disorder of the heart or blood vessels and included those who had experienced an acute myocardial infarction (MI), silent MI, coronary surgery, and/or strokes or who currently had atherosclerosis and/or a stent implants (*Hosein et al., 2020*; *Md Souza et al., 2012*; *Whitehead, Ford & Gama, 2014*; *WHO, 2016a*). Exclusion criteria included pregnant women and persons who had not resided in Trinidad and Tobago for the last 20 years. Non-CVD participants were recruited using flyers and were surveyed, measured, and tested at six locations throughout the country.

### Data collection

Ethical approvals were obtained from the University of Trinidad and Tobago Internal Review Board (UTTO/68/13) and the Ethics Committee of the Ministry of Health of

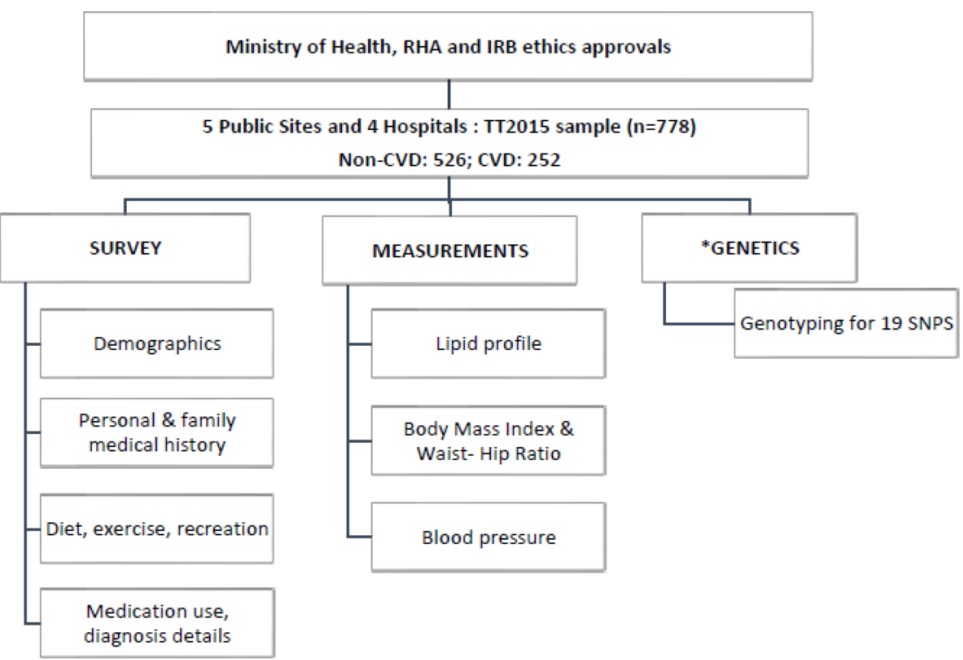

Figure 1  Flowchart illustrating the overall methodology and the parameters measured in this study.

Trinidad and Tobago to conduct this research. All participants received an overview of the study and its requirements and signed informed consent forms to acknowledge their voluntary participation.

For non-CVD participants, a questionnaire was administered to collect information on age, sex, ethnicity, smoking habits, family history of CVD, diet, and lifestyle. Measurements of weight, height, waist and hip circumference, blood pressure, and lipid profiles were recorded using a POC CardioChek PA analyser from PTS Diagnostics, Whitestown, IN, USA (*Hosein et al., 2020*; *Matteucci et al., 2014*; *Panz et al., 2005*) (Fig. 1).

For CVD participants, relevant data were collected from their hospital medical records at the first instance of a physician-diagnosed CVD event, which included any disorder of the heart or blood vessels. This group consisted of individuals who had experienced an acute myocardial infarction (MI), silent MI, coronary surgery, and/or strokes, or who currently had atherosclerosis and/or stent implants (*Md Souza et al., 2012*; *Whitehead, Ford & Gama, 2014*; *WHO, 2016a*). Data collected from these patients included their age, lipid profiles, blood pressure, pre-existing medical conditions, smoking habits, and BMI at the time of their first CVD event. Participants were also interviewed about their lifestyle at the time of their first CVD event.

Saliva swabs were taken from all participants for genetic analysis of single nucleotide polymorphisms (SNPs). Although this data yielded interesting independent information, it did not affect any of the models and is not discussed in this paper.

## Data analysis

Exploratory classification methods, outlined below, were used to develop algorithms to identify CVD status. The two most applicable models, which yielded algorithms of simple linear combinations of significant variables, were used to generate classification scores for all patients, just by substitution of the pertinent patient data into a model, shown as a generic Eq. (1):

$$Classification\ score = aA + bB + cC + \ldots\ldots\ldots\ldots + zZ \tag{1}$$

From overlapping histograms for the scores for known members of the non-CVD and CVD groups, it was possible to extract ranges which were populated only by non-CVD patients, only by CVD patients, and by a mix of both. This could be done, for each ethnic group, with each of the two new models developed and with the three well-known models.

All statistical analyses were carried out with SPSS V.22.

### CHAID classification model

The routine for a chi-square automatic interaction detector (CHAID) analysis was employed to estimate significant categorical and pseudo-categorical risk factors for CVD. A split sample approach, which randomly separated the sample into those cases used to build the model (Training) and those used to test it (Test), was used with a minimum number of parent nodes of 100 and child nodes of 50. The decision tree identified the classification variables for which there were significant chi-square tests for the branching at each node. The CHAID routine was performed three times (three models) to partition the data into statistically significant subgroups that were mutually exclusive and exhaustive. The split sample % was varied in these repeated analyses between 60% and 75% to check for maximize model classification.

### Discriminant analysis

Discriminant analysis (forward stepwise) was used to extract a classification model for the presence or absence of CVD in the TT2015 sample. The stepwise routine ensures that only significant variables are included in the final solution, which would be used to build the model. All assumptions of independence (non-collinearity), homoscedasticity, and multivariate normality were tested for residuals and for predictors. The Box's M test was used to estimate equality of covariance. Class means and standard deviations were compared to ensure that there were no patterns (increasing variances with increasing class means). Several analysis options for prior probabilities, selection of the type of covariance matrix, and resampling cross-validation classification techniques were explored in order to generate the model with the best classification results. The goodness of fit of the classification was estimated from the matrix of classification results. This gives the % of individuals correctly classified for both the original sample and those cases used

for cross-validation. The optimum classification model is that which gives the highest percentages of correctly classified original and cross-validated CVD and non-CVD cases.

### Logistic regression
Binary logistic regression (forward stepwise routine) was used to obtain a multivariate classification model with both categorical and continuous predictors. A classification matrix was generated to show the classification and misclassification percentages for each sub-group and the total sample. The Box-Tidwell procedure was used to test for linearity between the continuous independent variables and the logit.

### Data summaries and statistical tests of inference
Frequency distributions were used to summarize all the categorical demographic and health data. Responses for variables with 'check all that apply' options, such as those which captured information on co-morbidities or medications used, were ranked according to % calculated. Scale variables are described with summary statistics (means and standard deviations). Statistical tests of inference were used to estimate significant correlations (Chi-Square ($\chi$2) tests of independence) between pairs of categorical variables, usually a demographic and a variable denoting categorical health status, or to test the significance of impacts of demographic and behavioural categorical independent variables on scale health data responses (t and ANOVA).

### Receiver operating characteristic curves
Risk Scores were calculated for the local study sample using the existing risk models, QRISK2 2015 Batch processor (*Hippisley-Cox et al., 2007*), the Framingham Risk Calculator and the ASSIGN Risk Calculator (*Kannel, McGee & Gordon, 1976*; *Payne, 2012*). These scores, along with those from the two new models developed using discriminant analysis and logistic regression, were compared by generating their receiver-operating characteristic (ROC) curves, which quantify how well each model was able to discriminate between non-CVD and CVD participants. Area under the receiver-operating characteristic (AUROC) was used as a measure of the overall performance of a diagnostic test and was interpreted as the average value of sensitivity for all possible classification thresholds. In general, a diagnostic test with an AUROC value greater than 0.5 confirms that the model performs better than relying on pure chance and should have at least some ability to discriminate between subjects with and without a particular disease. Because sensitivity and specificity are independent of class prevalence, AUROC is also independent of any disparity in class (CVD *versus* non-CVD) sizes.

## RESULTS

### Sample description
The frequency distributions for the demographics of the study participants are shown in Table 1. Participants in the study had an overall mean age of 46.0 years ($\pm$12.8). Slightly more than half the sample was female (53%), most had secondary or higher-level education (81%), and almost half of the sample was single (48%). There was a relatively even distribution among the racial groups of Afro-Trinbagonians (35%), Indo-Trinbagonian

(32%), and Mixed-Trinbagonian (31%), closely reflecting the actual distribution of these groups in the larger population. Most were non-smokers (83%), with the remaining 17% being either a smoker or an ex-smoker. The TT2015 sample had a mean BMI of 26, with 53% classed as overweight/obese.

In Table 1, $p$ values are given for chi-square tests of independence (% frequency distribution categorical variables) or $t$-tests (means for continuous variables) comparing the non-CVD and CVD sub-groups. This is a preliminary exploration of potential CVD classification variables ($p < 0.05$). These are not definitive because some of these variables may be collinear and so in a linear model, only the more powerful classifier would be entered. Hence, some variables with significant tests here may not appear in the final models. It should be noted that repeated tests did not lead to specious correlation, requiring a Bonferroni-type adjustment of the $p$ values, as it could if repeated tests of significance were done for the same groups (*Frane, 2019*). Here, the disaggregated demographic, behavioural and comorbidity sub-groups in the different tests were different.

Comorbidities with CVD for the patients and similar conditions for the non-CVD individuals were recorded. Several, such as high blood pressure, high atrial fibrillation, chronic kidney disease and rheumatoid arthritis, were almost exclusively (89 to 93% of the total observed) in the CVD sub-group (Table 1). Others, like diabetes and high cholesterol, although not as prevalent overall, were also mainly in the CVD sub-group. The mean differences were all significant ($p < 0.05$). The effect is even more marked when considering that the non-CVD sub-group was twice the size of the CVD sub-group.

## Chi-square automatic interaction detector trees

Chi-square automatic interaction detector (CHAID) analysis was performed to identify the most important categorical predictors that were able to discriminate between non-CVD and CVD. Exhaustive CHAID, a modification to the basic CHAID algorithm, performs a more thorough merging and testing of predictor variables (*Kass, 1980*).

The CHAID routine was run on the sample data using different selection criteria to obtain optimum classification (Table 2). Model 3 was the most successful at predicting the CVD group in the TT2015 sample at 85% in the Training sample and 88% in the Test sample compared to Models 1 and 2 (Table 2). The results from the extraction of the significant classifying variables showed that the individual's age and the presence of high blood pressure were included in all three models (Table 3). Models 2 and 3 have the same three significant predictors but markedly different classification predictions in both the Training and Test sets (because of the selection criteria, including the splitting %, chosen).

Three major significant independent variables, the presence of high blood pressure, age in years and LDL levels were included in model 3. This model had an overall classification accuracy of 88.4% with its ability to discriminate between non-CVD and CVD participants. The tree analysis for Model 3 in Fig. 2 shows the 3-level CHAID tree with a total of eight nodes, of which five were terminal nodes. All individuals were divided into two subgroups from the root node to leaf nodes through different branches. The likelihood of an individual in this Trinidad and Tobago sample having a CVD varied from 0 to 32%. For example, 93% of persons with high blood pressure were likely to have a CVD incident. By using

**Table 1** Distribution of the major characteristics measured among the non-CVD and CVD participants for the sample population ($n - 778$).

| Parameter& Categories (Acceptable levels) | Total %/Mean (SD) | Non-CVD % Freq./Mean (SD) | CVD % Freq./ Mean (SD) | p |
|---|---|---|---|---|
| **Age (years)** | 46 (12.8) | 35 (14.1) | 56 (11.5) | 0.000 |
| 18–25% | 21.1 | 96.2 | 3.8 | |
| 26–41% | 29.7 | 90.1 | 9.9 | 0.000 |
| 42–56% | 24.1 | 54.1 | 45.9 | |
| 57–75% | 25.1 | 27.5 | 72.5 | |
| **Sex** | | | | 0.001 |
| Male % | 47.3 | 61.2 | 38.8 | |
| Female % | 52.7 | 73.1 | 26.9 | |
| **Ethnic group** | | | | 0.000 |
| African % | 34.6 | 84.8 | 15.2 | |
| East Indian % | 32.5 | 52.6 | 47.4 | |
| Mixed % | 31.0 | 64.8 | 35.2 | |
| Other % | 2.0 | 46.7 | 53.3 | |
| **Education Level** | | | | 0.305 |
| None/Primary % | 19.8 | 70.4 | 29.6 | |
| Secondary % | 49.6 | 65.0 | 35.0 | |
| Tertiary % | 30.5 | 70.1 | 29.9 | |
| **Marital status** | | | | 0.000 |
| Single % | 47.7 | 86.2 | 13.8 | |
| Married/Common-law % | 41.0 | 48.0 | 52.0 | |
| Divorced/Separated/Widowed % | 11.3 | 53.6 | 46.4 | |
| **Cigarette smoking** | | | | 0.000 |
| Never % | 82.9 | 71.2 | 28.8 | |
| Ex-smoker % | 8.9 | 9.8 | 90.2 | |
| Current smoker % | 8.2 | 75.8 | 24.2 | |
| **BMI (kg/m$^2$) (18.5 to 24.9)** | 26 (5.9) | 26 (6.2) | 26 (5.6) | 0.130 |
| **WHR (cm/cm)** | 0.86 (0.10) | 0.83 (0.09) | 0.94 (0.07) | $1.3 \times 10^{-50}$ |
| **Resting blood pressure (mmHg)** | | | | |
| Mean systolic (90–140 mmHg) | 124 (22.4) | 121 (19.9) | 126 (24.8) | 0.000 |
| Mean diastolic (60–90 mmHg) | 82 (13.1) | 82 (12.7) | 82 (13.4) | 0.995 |
| **Lipid profile (mg/dL)** | | | | |
| Total cholesterol (<200 mg/dL) | 166 (46.5) | 151 (37.0) | 181 (56.0) | 0.000 |
| LDL (<100 mg/dL) | 91 (25.1) | 78 (3.4) | 104 (46.8) | 0.000 |
| HDL (>40 mg/dL) | 47.5 (18.2) | 49 (17.8) | 46 (18.5) | 0.066 |
| Triglycerides (<150 mg/dL) | 130 (88.4) | 120 (82.9) | 139 (93.9) | 0.015 |
| **Co-morbidities** | | | | |
| High cholesterol % | 25.9 | 23.4 | 76.6 | 0.000 |
| High blood pressure (HBP) % | 21.1 | 6.7 | 93.3 | 0.000 |
| Diabetic % | 18.1 | 29.3 | 70.7 | 0.000 |
| Atrial fibrillation % | 8.7 | 7.7 | 92.3 | 0.000 |
| Chronic kidney disease % | 1.4 | 9.1 | 90.9 | 0.000 |
| Rheumatoid arthritis % | 1.2 | 11.1 | 88.9 | 0.000 |
**Table 1** (*continued*)

| Parameter& Categories (Acceptable levels) | Total %/Mean (SD) | Non-CVD % Freq./Mean (SD) | CVD % Freq./ Mean (SD) | p |
|---|---|---|---|---|
| **Family history** | | | | |
| Family history of diabetes % | 63.1 | 64.2 | 35.8 | 0.005 |
| Family history of HBP % | 62.5 | 63 | 37 | 0.000 |
| Family history of CVDs % | 38.6 | 37.8 | 62.2 | 0.000 |
| Family history of high cholesterol % | 24.4 | 64.7 | 35.3 | 0.152 |

**Table 2  CHAID split sample for the three models with percentage correctly predicted for the CVD group and mean overall correctly predicted percentage.**

| | % correct predicted for CVD group | | |
|---|---|---|---|
| | Training | Test | Mean overall (Non-CVD and CVD) |
| MODEL 1 | 59.3 | 60.9 | 85.8 |
| MODEL 2 | 69.0 | 54.8 | 85.9 |
| MODEL 3 | 85.2 | 89.4 | 88.4 |

this type of decision tree model, researchers can identify the combinations of factors that constitute the highest (or lowest) risk for a condition of interest.

## Logistic regression

A forward stepwise binary logistic regression was performed for the TT2015 sample on nineteen variables. The final logistic regression model included ten predictors of the odds of an individual in this sample having CVD (Table 4). The results showed that the odds increased as people age, for males, for married individuals (compared to single, including separated, divorced, and widowed), for smokers, for those with low TC/HDL and high LDL levels (mg/dL), high cholesterol, atrial fibrillation, high blood pressure and a family history of CVD. The model explained 55% (Cox and Snell $R^2 = 0.55$) to 86.2% (Nagelkerke R2 =0.862) of the variance. Ninety-nine percent (99%) of the non-CVD participants were correctly classified while 88% of the CVD participants were correctly classified using this model.

## Classification of logistic regression model risk scores

The sample scores were computed using the logistic regression model (LinRisk), as detailed in Eq. (1). A stacked histogram of these scores was plotted using different colours for the non-CVD and CVD groups (Fig. 3). Using this histogram, two classification ranges were marked off to represent scores which predominantly classify non-CVD and CVD. A third range was identified in which there was clear overlap between scores for individuals with and without CVD. This last was named the non-differentiating range. These ranges were defined as 'clearly non-CVD' (−12.0 to −4.01), 'non-differentiating range' (−4.00 to 0.00) and 'clearly CVD' (0.01 to 12.0). This non-differentiating range of scores essentially captures the misclassification cases, which include a mix of non-CVD and CVD participants. Better

**Table 3  Summary of the three CHAID tress developed.**

| | | MODEL 1 | MODEL 2 | MODEL 3 |
|---|---|---|---|---|
| Specifications | Growing method | Exhaustive CHAID | Wxhaustive CHAID | Exhaustive CHAID |
| | Dependent Variable | Case Control | Case Control | Case Control |
| | Independent Variables | Age, Sex, Ethnicity, Marital_status, Highest_Edu, Smoking, BMI, Tot.Chol, Triglycerides, LDL, HDL, TC_HDL, SBP, DBP, Diabetic, High_Cholesterol, Chronic_ Kidney_Disease, Atrial_fibrillation, Rheumatoid_arthritis, HBP, Family_HBP, Family_High_Cholest., Family_Diabetes Family_CVD | Age, Sex, Ethnicity, Marital_status, Highest_Edu, Smoking, BMI, Tot.Chol, Triglycerides, LDL, HDL, TC_HDL, SBP, DBP, Diabetic, High_Cholesterol, Chronic_ Kidney_Disease, Atrial_fibrillation, Rheumatoid_arthritis, HBP, Family_HBP, Family_High_Cholest., Family_Diabetes Family_CVD | Age, Sex, Ethnicity, Marital_status, Highest_Edu, Smoking, BMI, Tot.Chol, Triglycerides, LDL, HDL, TC_HDL, SBP, DBP, Diabetic, High_Cholesterol, Chronic_ Kidney_Disease, Atrial_fibrillation, Rheumatoid_arthritis, HBP, Family_HBP, Family_High_Cholest., Family_Diabetes Family_CVD |
| | Validation | Split Sample | Split Sample | Split Sample |
| | Maximum Tree Depth | 3 | 3 | 3 |
| | Minimum Cases in Parent Node | 100 | 100 | 100 |
| | Minimum Cases in Child Node | 50 | 50 | 50 |
| Results | Independent Variables Included | HBP, Age, Ethnicity | HBP, LDL, Age | HBP, LDL, Age |
| | Number of Nodes | 8 | 7 | 7 |
| | Number of Terminal Nodes | 5 | 4 | 4 |
| | Depth | 3 | 3 | 3 |

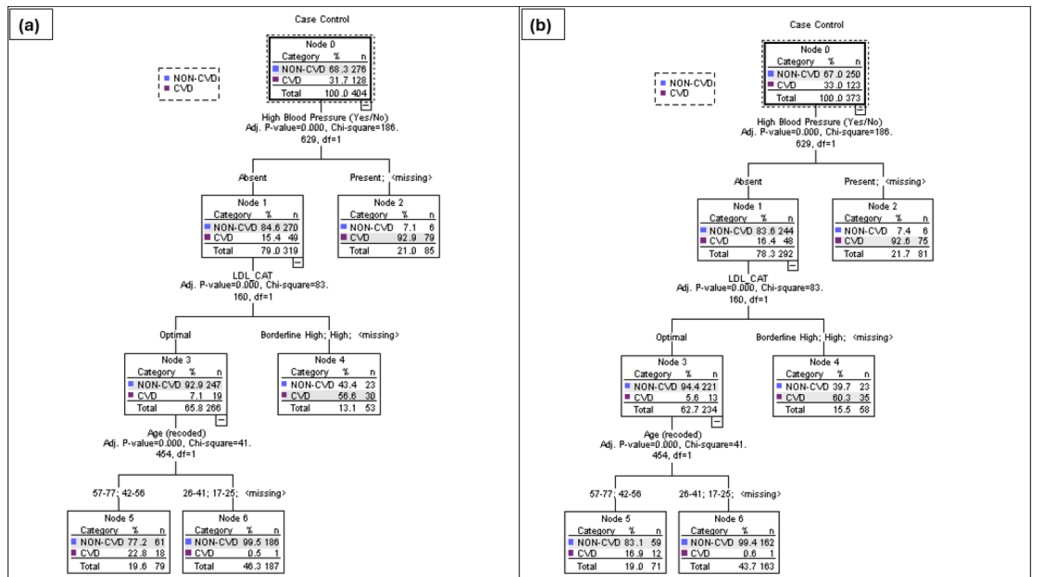

**Figure 2  CHAID tree diagram for Training (A) and Test (B) split sample for Model 3, using the presence of high blood pressure, LDL levels (mg/dL) and age range in years.**

**Table 4** Table showing significant variables in a binary logistic regression analysis (forward stepwise) estimation for the odds of having CVD in the Trinidad and Tobago sample.

| Significant Variables | Wald | Sig. |
|---|---|---|
| Age until first CVD event or current (years) | 21.81 | .000 |
| Sex (Female) | 4.64 | .031 |
| Marital status (Married >single or Divorced/Separated/Widowed) | 4.43 | .035 |
| Smoking (Smoker >non-smoker, ex-smoker) | 6.99 | .008 |
| TC/HDL ratio | 6.72 | .010 |
| LDL (mg/dL) | 9.90 | .002 |
| High cholesterol (Present) | 12.64 | .000 |
| Atrial fibrillation (Present) | 4.57 | .033 |
| High blood pressure (Present) | 30.00 | .000 |
| History of family CVD (Present) | 13.14 | .000 |
| Constant | 33.69 | .000 |

models are those with narrower ranges of misclassification and smaller ratios of CVD/ non-CVD participants in this non-differentiating class because the risk to a misclassified CVD individual is far greater than that to a misclassified non-CVD person so that evaluation of model performance must recognize this. If a non-CVD individual is misclassified and is given a diagnosis of CVD from a medical practitioner using a risk classification model for guidance, it would usually be because their risk score is on the borderline between classes and what follows should be relatively benign. For them, prescribed alterations to their diet and exercise routines- usually the first course of action- can only help not hurt. On the other hand, someone who should truly be in the CVD risk score range, who is not identified as such, will not take precautions or medication, probably exacerbating his or her condition.

## Discriminant analysis

Stepwise discriminant analysis (DA) was performed for the TT2015 sample to test the significance of nineteen variables which could be included in the optimal classification algorithm. Of these, only 13 significantly discriminating factors ($p < 0.05$) were extracted (Table 5). According to the discriminant analysis model (DiscrimRisk), the scores increased as people age in general but more for males, married individuals, ex-smokers, those with lower ratios of TC/HDL but higher low-density lipoprotein (LDL) levels (mg/dL), with higher blood pressures (systolic but not diastolic), with high cholesterol, atrial fibrillation and a family history of CVD. These, except for atrial fibrillation and the breakdown into both systolic and diastolic blood pressures, are common to the LinRisk model, albeit with different weights given to the different variables. The discriminant analysis was different from the LinRisk model by identifying the role of education. Increasing education increases CVD risk. On the surface, this seems counterintuitive, especially if an argument is used that knowledge and awareness are risk controls (*Croquelois & Bogousslavsky, 2006*). Although Trinidad and Tobago is listed as a high-income country (*Carneiro et al., 2014*), the sample suggests that educated and high-income individuals are more susceptible to

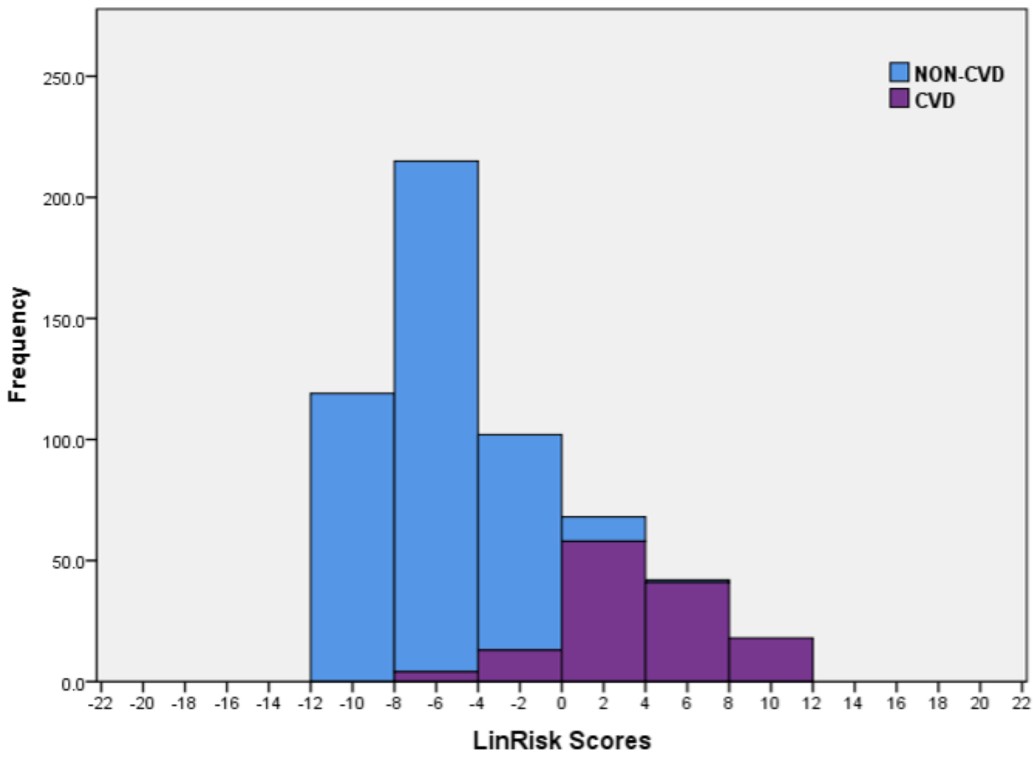

**Figure 3** Histogram of the LinRisk scores from the logistic regression model of 10 significant predictor variables from the TT2015 sample for non-CVD and CVD participants.

CVD risk factors like obesity because of their access to surplus/excess food and a lower level of engagement in manual labour-intensive occupations which are typically associated low-income countries (*Dinsa et al., 2012*).

Box's M was significant ($p < 0.000$) with dissimilar log determinants (3.9 for the non-CVD group and 7.3 for the CVD group) which indicates a violation of the assumption of equal covariance matrices. This test was overly sensitive to situations in which multivariate normality does not hold. It was also overly sensitive to small and large sample sizes, with low power with the former (false non-significance) and too much of a tendency to find significant effects with larger samples so it is often recommended for larger samples that the test should be used at a significance level $p < 0.001$. However, if the group sizes are unequal, as in this case, and a significant inference was made even at the lower $p$-value, then the test was not considered robust (*Tabachnic & Fidell, 2001*). A non-robust significant test of an assumption in this analysis means the Discriminant function obtained will be susceptible to greater classification errors.

The discriminant function revealed a significant association between groups and all predictors (canonical function coefficient = 0.853). It explained 73% of the variance among the values of the significant variables for all the cases in the analysis (Wilks Lambda = 0.273). For the original and cross-validated cases, 98% of the non-CVD participants

**Table 5** Significant predictors in the discriminant function for classification of non-CVD and CVD participants in the TT2015 sample.

| Variables | Wilks Lamda | P value |
|---|---|---|
| Age (years) | 0.297 | 0.000 |
| Sex (Male/Female) | 0.278 | 0.000 |
| Secondary Education Level | 0.278 | 0.000 |
| Tertiary Education Level | 0.278 | 0.000 |
| Married | | 0.000 |
| Ex-Smoker | 0.283 | 0.000 |
| TC/HDL ratio (mg/dL) | 0.277 | 0.000 |
| LDL (mg/dL) | 0.282 | 0.000 |
| Systolic blood pressure (mmHg) | 0.276 | 0.000 |
| Diastolic blood pressure (mmHg) | 0.284 | 0.000 |
| High Cholesterol (Yes/No) | 0.280 | 0.000 |
| Atrial fibrillation (Yes/No) | 0.280 | 0.000 |
| High Blood Pressure (Yes/No) | 0.371 | 0.000 |
| Family history of CVD (Yes/No) | 0.285 | 0.000 |
| Constant | | |

were correctly assigned while 87% of the CVD participants were correctly classified using this model. Of the original grouped cases, 96% overall were correctly classified.

Similar to the LinRisk model above, a stacked histogram of these scores was plotted using different colours for the non-CVD and CVD groups (Fig. 4). It was used to classify 'Clearly non-CVD' ($<-0.25$), 'non-differentiating range' ($-0.25$ to $0.90$) and 'Clearly CVD' ($>0.90$). The non-differentiating range of scores captured the misclassification cases, which included a mix of non-CVD and CVD participants.

## Summary of the three developed CVD risk prediction models

The three models, tested for discrimination of non-CVD and CVD participants in the TT2015 sample, were obtained from a CHAID decision tree, Logistic Regression and Discriminant Analysis. All three models (Table 6) included the three predictors for CVD, first obtained from the least rigorous CHAID analysis, namely the individual's high blood pressure (HBP), levels of low-density lipoprotein (mg/dL) and age in years (at time of first CVD incident or current). Note that although these are all classification models, the LinRisk and DiscrimRisk models can predict the class assignments of new individuals. This is seen from the classification power of the Test sets obtained using the algorithms extracted with the Training sets in both the LinRisk and DiscrimRisk models. They can be considered 'current' Risk prediction models, even though obtained from cross-sectional analyses. It was interesting to note that Model 1, obtained from the CHAID analysis, although not as successful as Models 2 &3, recognised ethnicity as an important predictor of CVD, while the other two did not even though that variable was tested in both analyses (Table 6). This may be due to ethnicity being collinear with a more powerful classifier/predictor so that it is left out of any model obtained using a stepwise approach to goodness of
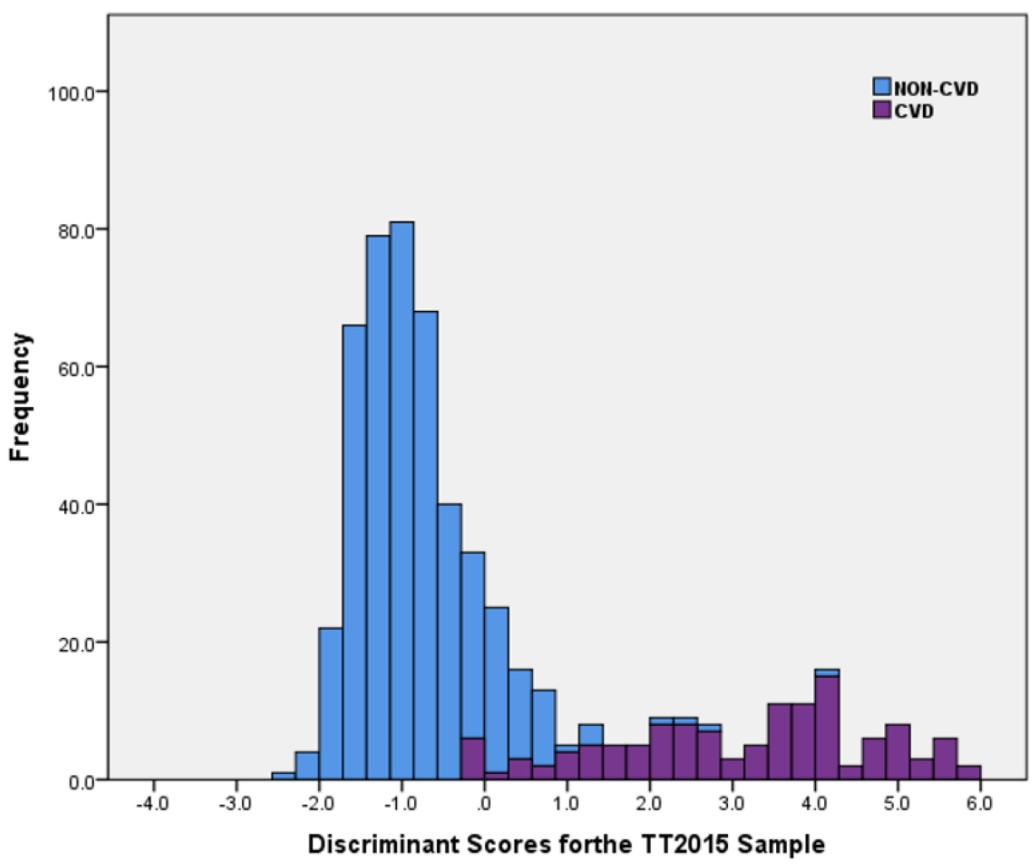

**Figure 4** Histogram of discriminant scores from the discriminant analysis model (DiscrimRisk) using significant predictor variables in the TT2015 sample for non-CVD and CVD participants.

fit/explanatory/classification power of the final algorithm. Of the three models, the logistic regression (non-CVD: 99% and CVD: 88%) and discriminant analysis (non-CVD: 98% and CVD: 87%) were not just better able to correctly classify non-CVD and CVD participants compared to the CHAID model, they also were able to identify three numerical ranges for class assignment, depending on the scores of individuals. (Table 7).

## Comparison of variables in all six CVD risk models

The six models for a CVD classification were the ASSIGN, Framingham (2008), QRISK2, CHAID, logistic regression (LG) and discriminant analysis (DA). Note that the first three models mentioned were summarized and compared in detail in a previous publication (*Hosein et al., 2020*). They are repeated here to compare their CD risk prediction performance with the proprietary models developed in this study, LinRisk and DiscrimRisk. These models, used in North American and European countries, estimate the risk of a person having a CVD in the next 10 years (*D'Agostino et al., 2008*; *Hippisley-Cox et al., 2007*; *Woodward, Brindle & Tunstall-Pedoe, 2007*). The LinRisk and DiscrimRisk models predict the risk of an unknown individual to be assigned correctly to one of three classes, confirmed CVD, confirmed non-CVD, and indeterminate requiring further tests.

**Table 6   Variables used in the six CVD risk models for the TT2015 sample.**

| Predictor | LinRisk | DiscrimRisk | CHAID | Framingham | ASSIGN | QRISK2 |
|---|:---:|:---:|:---:|:---:|:---:|:---:|
| Age (years) | ✓ | ✓ | ✓ | ✓ | ✓ | ✓ |
| Presence of HBP | ✓ | ✓ | ✓ | | | ✓ |
| Family history of CVD | ✓ | ✓ | | ✓ | ✓ | ✓ |
| Presence of High Cholesterol | ✓ | ✓ | | | | |
| Presence of Atrial Fibrillation | ✓ | ✓ | | | | ✓ |
| Sex | ✓ | ✓ | | ✓ | ✓ | ✓ |
| Smoking | ✓ | ✓ | | ✓ | ✓ | ✓ |
| TC/HDL | ✓ | ✓ | | ✓ | ✓ | ✓ |
| Marital status | ✓ | ✓ | | | | |
| LDL | ✓ | ✓ | ✓ | | | |
| Diabetic | | | | ✓ | ✓ | ✓ |
| Diastolic BP | | ✓ | | | | ✓ |
| No. of cigarettes/day | | | | | ✓ | ✓ |
| Presence of Left Ventricular Hypertrophy | | | | | ✓ | |
| Social deprivation[a] | | | | | ✓ | ✓ |
| Systolic BP | | ✓ | | ✓ | ✓ | ✓ |
| BMI | | | | | | ✓ |
| Presence of Rheumatoid arthritis | | | | | | ✓ |
| Presence of chronic kidney disease | | | | | | ✓ |
| Ethnicity | | | | | | ✓ |
| Education Level | | ✓ | | | | |
| Total number of predictors in the model | 10 | 13 | 3 | 7 | 10 | 14 |

**Notes.**
[a]Data for this predictor unavailable for all models.

**Table 7   Classification characteristics for five CVD risk prediction models evaluated.**

| Characteristic | Established models | | | New models | |
|---|---|---|---|---|---|
| | Assign | Fram.[a] | QRISK2 | LinRisk[b] | DiscrimRisk[c] |
| Sensitivity | 49.6 | 44.7 | 62.1 | 86.0 | 89.7 |
| Specificity | 80.3 | 87.4 | 86.4 | 76.4 | 81.9 |
| Positive predictability | 77.9 | 77.8 | 87.0 | 91.4 | 99.0 |
| Negative predictability | 87.1 | 92.9 | 92.3 | 98.2 | 96.0 |
| Non-Differentiating risk score range | 5.00 to 12.50 | 10.00 to 22.50 | 5.00 to 15.00 | −4.00 to 0.00 | −0.25 to 0.90 |
| % Total sample in Non-differentiating range | 16.84 | 14.04 | 13.50 | 18.2 | 14.7 |
| Relative Risk of CVD patient being in non-differentiating range | 1.99 | 3.74 | 2.58 | 0.46 | 0.59 |

**Notes.**
[a]Framingham (FRAM.) CVD risk model (2008).
[b]Logistic regression (LinRisk) CVD model.
[c]Discriminant analysis (DiscrimRisk) CVD model.

The QRISK2 model uses the most predictors (14), followed closely by our proprietary DiscrimRisk model (13) while CHAID Analysis provides classification with just three

predictors (Table 6). In the final performance assessment, it will be important to consider that the more measurements needed to estimate an individual's risk score, the more time-consuming and expensive it is. The connection, if any, between the number of predictors and the efficacy of the models will be discussed.

Age was a common predictor for all six models. The presence of a family history of CVD, sex, smoking status and the ratio of total cholesterol to high-density lipoprotein (HDL), were predictors in all models except that from CHAID. All 10 predictors in the LinRisk model were also included in DiscrimRisk (Table 6). The latter is a bigger algorithm, including the Education level, the only one of the six models which does, and specifically diastolic and systolic BP levels. Information on these extra three variables in the DiscrimRisk model should be readily obtainable. Notably, all three international models include systolic BP but only the QRISK2 prediction model also includes diastolic BP.

The presence of diabetes was a predictor for the three established CVD risk models but was not recognised as a predictor in the three models developed in this study. This could be due to one or more reasons. It is possible that the other demographics, which are included in the three proprietary models, provide the same information and more than the Diabetes status of the individual, in which case collinearity would stop its inclusion in either LinRisk or DiscrimRisk, both of which utilized stepwise procedures. (Whether or not it could be included in a CHAID model could depend on the random splitting of the sample into Training and Test sets. This is the issue which could make this less reliable than LinRisk and DiscrimRisk.) Another possible consideration is that Diabetes is so unfortunately prevalent in the Trinidad and Tobago population that it is not sufficiently discriminatory between individuals with different CVD statuses. This is explored later in this paper when an indirect association between ethnicity and the sensitivity of the models is examined, particularly as Diabetes is more markedly prevalent among Indo-Trinbagonians.

The inclusion of a social deprivation score as a predictor for ASSIGN and QRISK2 was not possible for any of the six models because it does not exist for the Trinidad and Tobago population. It is expected that the urban intensity index recently published for Trinidad and Tobago will be evaluated for use as such a factor in the near future (*Chadee & Stoute, 2018*).

## DISCUSSION

### Performance assessment of five CVD risk prediction models

The CVD risk scores produced by the QRISK2, Framingham and ASSIGN models estimate the likelihood of having a CVD event in the next 10 years and were developed from cohort studies by evaluating a sample population over many decades. Our proprietary models, LinRisk and DiscrimRisk, recognise individuals who are in CVD status currently. For individuals, who may be at immediate risk of their first CVD incident, that difference could be lifesaving. Additionally, because the risk score is estimated, individuals with higher scores in the non-CVD region and all individuals with scores in the non-differentiating range can be alerted to make the necessary lifestyle changes, which could forestall the progression of the disease. The CHAID model does not provide a risk score for individuals, and as such was not evaluated in the comparative analysis of the CVD risk models.

Sensitivity and specificity are terms used to evaluate a clinical test. In this study, these metrics are applied to the five CVD risk prediction models, with sensitivity defined as the ability of the model to correctly 'detect' individuals in the CVD group and specificity as the level of detection of those in the non-CVD group (*Lalkhen & McCluskey, 2008*). Both metrics are independent of the population of interest (*i.e.* TT2015). In addition to these, positive and negative predictabilities are other metrics used with measures (tests or models) of evaluation of disease status (*Akobeng, 2007*; *Lalkhen & McCluskey, 2008*). In this study, for each of the models, the positive predictability measures the percentage of all those identified as CVD patients who truly had that status while the negative predictability is the percentage of individuals, who truly had never had a CVD event and were correctly assigned by the model to the non-CVD group. These operational definitions follow the epidemiology test standards. (*Parikh et al., 2008*).

The last three rows in Table 7 give more model performance evaluation information. There is the breadth of the non-differentiating range of scores, the % of the total sample which falls in this range, and the relative risk of CVD individuals in that range (% Total CVD group in range/% Total non-CVD group in range). In every one of these last three statistics, smaller values are better.

The two proprietary models outperform the international standards on most of the metrics but in particular for sensitivity, for which they were 27% to 45% more sensitive. High sensitivity is important where the test (multivariate risk score from the models) is used to identify a serious but treatable disease such as a CVD (*Lalkhen & McCluskey, 2008*). The LinRisk and DiscrimRisk models also have comparable specificities with the other three, better positive and negative predictabilities, relatively smaller (and so better) non-differentiating ranges, and comparable but slightly larger values of sample % in the non-differentiating range. The proprietary models had noticeably lower relative risks of a CVD patient being assigned to that range than did the international models. The odds of a CVD patient, compared to a non-CVD individual, being assigned to the non-differentiating range were much higher for the ASSIGN (1.99), Framingham (3.74), and QRISK2 (2.58) than those for LinRisk (0.46) and DiscrimRisk (0.59). This indicates that these proprietary models were more likely than the established models to include non-CVD rather than CVD individuals in the non-differentiating ranges, which is the preferable outcome. This is an important consideration as mentioned earlier since misclassifying a CVD patient has far worse health implications than incorrectly identifying the status of an individual without the disease.

It is not as clear-cut to choose between the two proprietary models in terms of performance. DiscrimRisk has better sensitivity, specificity, positive predictability, a smaller non-differentiating range and a smaller percentage of individuals in the total sample with scores falling into the non-differentiating range (14.7% compared with 18.2%). For the other two metrics, negative predictability, and relative risk of an individual with CVD, registering a score in the non-differentiating range, the two models are quite comparable. Finally, although DiscrimRisk is a longer algorithm with three extra variables, compared to LinRisk, these (education and blood pressure readings) are easily and cheaply gathered or

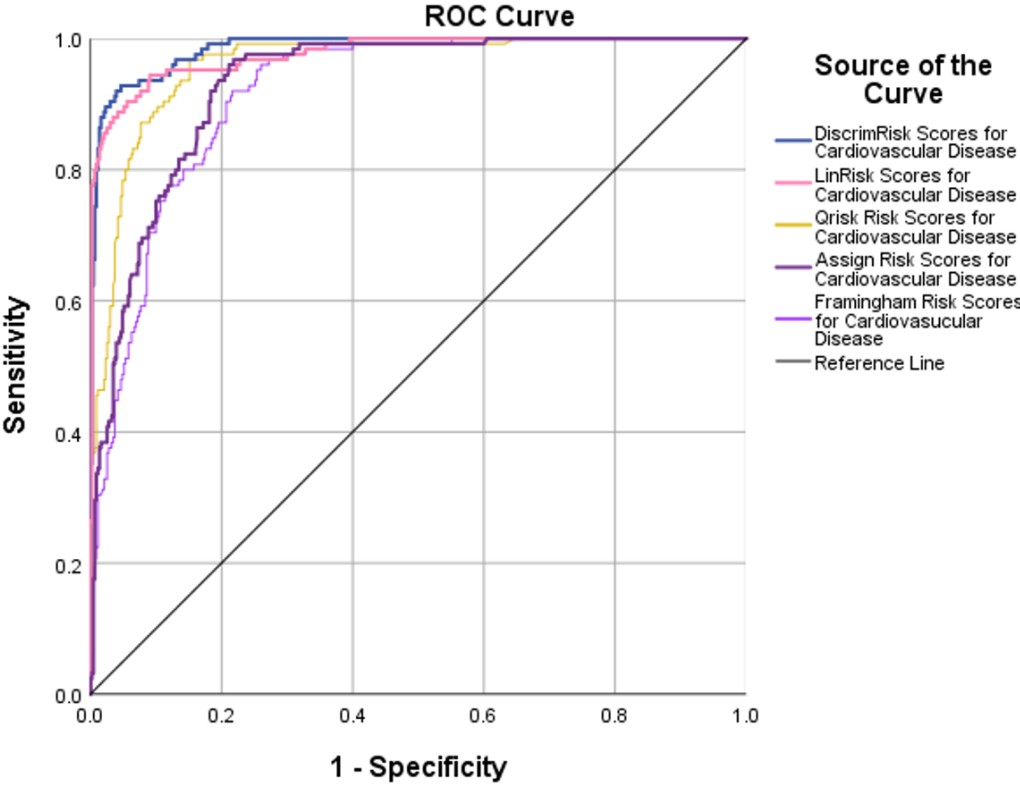

**Figure 5** ROC curves with AUROC values from the five risk models evaluated to discriminate between non-CVD and CVD participants in the TT2015 sample.

measured, worth collecting in order to get a better performance. That said, as is discussed later, using both models in tandem may prove to be the best approach for some individuals.

## Receiver operator characteristic curves

Receiver operator characteristic (ROC) curves are a plot of false true positives against true false positives for all cut-off values. The area under the curve (AUC) of a perfect test is 1.0 and that of a useless test, no better than tossing a coin, is 0.5. ROC curves for 5 of the models were plotted (Fig. 5). The discriminant analysis performed the best. It has an AUC = 0.986, with a 95% upper confidence limit (UCL) for this area of 99%. The Framingham risk model performed the poorest (AUROC =0.916, largest C.I. =0.046). The results show that the DiscrimRisk and LinRisk models performed much better for the TT2015 sample than any of the ASSIGN, Framingham, or QRISK2 models.

## Association of ethnicity with CVD assignment to the non-differentiating range

In this study, the ethnic distribution of the participants was 34.6% Afro-Trinbagonian, 32.4% Indo-Trinbagonian, 31.0% Mixed-Trinbagonian (African and East Indian) and 2.0% Other ethnicities. This almost exactly matches the distribution in the larger population,

**Table 8  Discrimination power of each of the five models tested in this study by ethnicity.**

| Ethnicity | Category | Established CVD risk mod | | | New models | |
|---|---|---|---|---|---|---|
| | | ARS | FRS | QRS | LinR | Discr. |
| Afro-Trinbagonian | Non-CVD | 81.2 | 87.9 | 90.6 | 76.2 | 84.8 |
| | Non-differentiating –Non-CVD | 13.0 | 7.0 | 6.7 | 21.8 | 13.7 |
| | Non-differentiating - CVD | 20.0 | 33.3 | 30.0 | 8.3 | 22.7 |
| | CVD | 50.0 | 50.0 | 52.5 | 79.2 | 77.3 |
| Indo-Trinbagonian | Non-CVD | 80.0 | 88.3 | 81.5 | 75.9 | 81.8 |
| | Non-differentiating –Non-CVD | 13.1 | 10.0 | 10.0 | 23.1 | 17.3 |
| | Non-differentiating - CVD | 23.9 | 31.1 | 12.8 | 8.8 | 7.7 |
| | CVD | 49.6 | 45.9 | 71.8 | 87.7 | 90.4 |
| Mixed-Trinbagonian | Non-CVD | 80.4 | 87.5 | 85.6 | 76.1 | 78.2 |
| | Non-differentiating –Non-CVD | 11.1 | 9.6 | 10.5 | 18.8 | 18.6 |
| | Non-differentiating - CVD | 31.3 | 36.5 | 33.7 | 9.8 | 4.2 |
| | CVD | 50.6 | 38.5 | 54.2 | 88.2 | 95.8 |

**Notes.**

ARS, ASSIGN Risk Score Model; FRS, Framingham Risk Score Model; QRS, QRISKII Risk Score Model; LinR, LinRisk Model; Discr., DiscrimRisk Model.

offering a good opportunity for a more granular study of ethnicity and the classification efficacy of models, especially in the non-differentiating range. Further, this strengthens the potential generalizability of the inferences from this study to the population as a whole. As is discussed below, this approach yielded much more information than the inclusion of ethnicity as a predictor in a risk model. It was not significant in either of the proprietary models and was only included in QRISK2 of the three established models.

For comparative performance assessments among the five models, in terms of classification accuracy for different ethnic groups, their classification percentage frequency distributions for all participants in the TT2015 sample were disaggregated by Ethnicity and then by four sub-categories, as clearly non-CVD, non-CVD persons in the non-differentiating range, CVD patients in the non-differentiating range and clearly CVD. Interestingly, some individuals can be completely misclassified, so that a CVD individual may be recognized as clearly non-CVD and vice versa. These individuals are not in the non-differentiating range. This accounts for instances in Table 8, where the percentages that should add up to 100% do not do so. The percentages for the cross-tabulations are shown in Table 8. Note these values were extracted from individual cross-tabulations for each of the five models so the row percentages do not add up to 100. In the columns, both (CVD + non-differentiating CVD) and (non-CVD + non-differentiating non-CVD) for each ethnic group should add up to 100%. Any total less than 100% represents the % of individuals completely misclassified as being clearly in the other group. This is important to note to assess performance. This is important in assessing performance.

For the DiscrimRisk model, the totals (%CVD + % non-differentiating CVD OR % non-CVD +% non-differentiating non-CVD) are never less than 96%, with most at 99% or 100%, regardless of ethnicity, suggesting that this model only completely misclassified very small percentages of CVD or non-CVD individuals. The LinRisk model has more of

**Table 9  Misclassification's ratios and percentages for five risk models.**

| Ethnicity | Category | Established CVD Risk Mod | | | New Models | |
|---|---|---|---|---|---|---|
| | | ARS | FRS | QRS | LinR | Discr. |
| Afro-Trinbagonians | CVD/non-CVD in non-Differentiating range | 1.54 | 4.76 | 4.48 | 0.38 | 1.66 |
| | non-CVD classified as CVD (%) | 5.8 | 5.1 | 2.7 | 2.0 | 1.5 |
| | CVD classified as non-CVD (%) | 30.0 | 16.7 | 17.5 | 12.5 | 0 |
| Indo-Trinbagonians | CVD/non-CVD in non-Differentiating range | 1.82 | 3.11 | 1.28 | 0.38 | 0.45 |
| | non-CVD classified as CVD (%) | 6.9 | 1.7 | 8.5 | 1.0 | 0.9 |
| | CVD classified as non-CVD (%) | 26.5 | 23.0 | 5.4 | 3.5 | 1.9 |
| Mixed-Trinbagonians | CVD/non-CVD in non-Differentiating range | 2.82 | 3.80 | 3.21 | 0.52 | 0.23 |
| | non-CVD classified as CVD (%) | 8.5 | 2.9 | 3.9 | 5.1 | 3.2 |
| | CVD classified as non-CVD (%) | 18.0 | 25.0 | 22.1 | 2.0 | 0 |

**Notes.**

ARS, ASSIGN Risk Score Model; FRS, Framingham Risk Score Model; QRS, QRISKII Risk Score Model; LinR, LinRisk Model; Discr., DiscrimRisk Model.

these totals deviating from 100%, with the lowest being 87.5% (8.3% non-differentiating CVD and 79.2% CVD) for Afro-Trinbagonians, suggesting that 12.5% of CVD patients in this ethnic group were misclassified as clearly non-CVD by the LinRisk model. Every other total for this model for the other two ethnic groups is 95% or higher. The other three models all completely misclassified significant percentages of CVD individuals, the worst cases being for Afro- and Indo-Trinbagonians for the Assign model. For these, the totals are only 70% and 73.5% respectively for CVD individuals classified either as clearly CVD or in the non-differentiating range. This means that this model misclassified 30% of Afro- and 26.5% of Indo-Trinbagonians with CVD as having a non-CVD status.

Another aspect of the models' classification performances is captured in Table 9. In this table are listed the ratios of CVD/non-CVD in the model's non-differentiating ranges for the three ethnic groups and the % of CVD patients and % non-CVD individuals, each classified as clearly belonging to the other group. Of particular interest are the ratios and the % CVD classified as non-CVD, which is a more severe misclassification than assignment to the non-differentiating class. This is actually (1- Sensitivity) for each ethnic group.

The DiscrimRisk model classifies either NO (0% for Afro- and Mixed-Trinbagonians) or FEW (1.9% for Indo-Trinbagonians) CVD patients as being in the non-CVD group –Table 9. It is the best performer in this most serious misclassification. It is followed by the LinRisk model (12.5% for Afro-Trinbagonians, 3% for Indo-Trinbagonians, and 2% for the mixed Trinbagonians). The established models are very poor predictors using this metric. Of these, QRISK2, was the only model which incorporates ethnicity in its algorithm is a clear best (5.4% compared to 23% for the Framingham and 26% for the Assign models) for Indo-Trinbagonians and a close second best for Afro-Trinbagonians (17.5% compared to 16.7% for the Framingham model) and for Mixed-Trinbagonians (22.1% compared to 18% for the ASSIGN model). Only for the Mixed-Trinbagonians is the Assign model the best. It is the worst model for the other ethnic groups.

Performance assessment based on the CVD/non-CVD ratio in the non-differentiating range again shows a marked and significantly ($p < 0.05$) better performance for the

two proprietary models than for the established models. They are the best for all ethnic groups. Only the Assign model for Afro-Trinbagonians is comparable to the result with the DiscrimRisk model. LinRisk is the best performer, with the lowest ratio for Afro (0.38) and Indo-Trinbagonians, with DiscrimRisk a close third (1.66 for Afro-) or second (0.45 for Indo-Trinbagonians. For Mixed-Trinbagonians, however, DiscrimRisk has the lowest relative risk (0.23) with LinRisk a close second (0.52). The established models have relative risks from 1.54 to 4.76. The Assign model has the smallest ratios among those for the established models for Afro-Trinbagonians (1.54) and Mixed-Trinidadians (2.82). The Framingham had the largest ratios of CVD/non-CVD in the non-differentiating range for all models.

Even though Ethnicity was not a significant predictor in either LinRisk or DiscrimRisk, it is clear from the examination at this micro level that these models gave significantly different results ($p < 0.05$) to individuals in the different ethnic groups from the two-classification metrics examined in detail above. The DiscrimRisk model worked best for Mixed-Trinbagonians and as good if not better than LinRisk for the other ethnic groups.

### Misclassification example: CD707

Participant CD707 is a Trinidadian male, 40 years old, of mixed ethnicity, married, with a tertiary education. He has high cholesterol (TC =224, HDL =56, LDL =144, TG =122 mg/dL), high blood pressure (135/79), overweight (BMI =29.2), but with no history of smoking, diabetes, chronic kidney disease, rheumatoid arthritis or family CVD.

The three established models identified participant CD707 as Low Risk (Fig. 6) with a less than 10% chance of having a CVD event in the next 10 years (ASSIGN RS =4.23%, Framingham RS =4.42%, and QRISK2 RS =2.79%). Both DiscrimRisk (RS = 3.63) and LinRisk (RS = 1.76) identified the participant as clearly CVD, as can be seen from where these scores are classed in Figs. 3 and 2, respectively.

Participant CD707 had a myocardial infarction (MI) and was diagnosed with multi-vessel coronary artery disease (CAD) requiring three Percutaneous Coronary Interventions (PCI, formerly known as angioplasty with stents). Each PCI can cost up to TT$120,000.00 (US$18,750.00) per procedure (*Hassanali, 2014*). The cost of treating CVD increases each year in Trinidad and Tobago and was last estimated at 6 billion at the end of 2016, according to Trinidad and Tobago's Health Minister, Terrence Deyalsingh (*Paul, A-L, 2016*). The ability to estimate a person's risk of having a CVD is important towards prevention and treatment which consequently contributes towards lowering the national healthcare fiscal burden. For CD707, the established models (Framingham, ASSIGN, and QRISK2) cannot be used for classification since they are under-estimating CVD risk compared to the proprietary models (DiscrimRisk and LinRisk).

### CONCLUSION

The overall high sensitivity, high positive predictability, low relative risk of a CVD participant being in the non-differentiating range, and a high percentage of correctly classified CVD and non-CVD individuals make the DiscrimRisk model the best at CVD discrimination for the TT2015 sample. This is crucial for the measurement, monitoring,
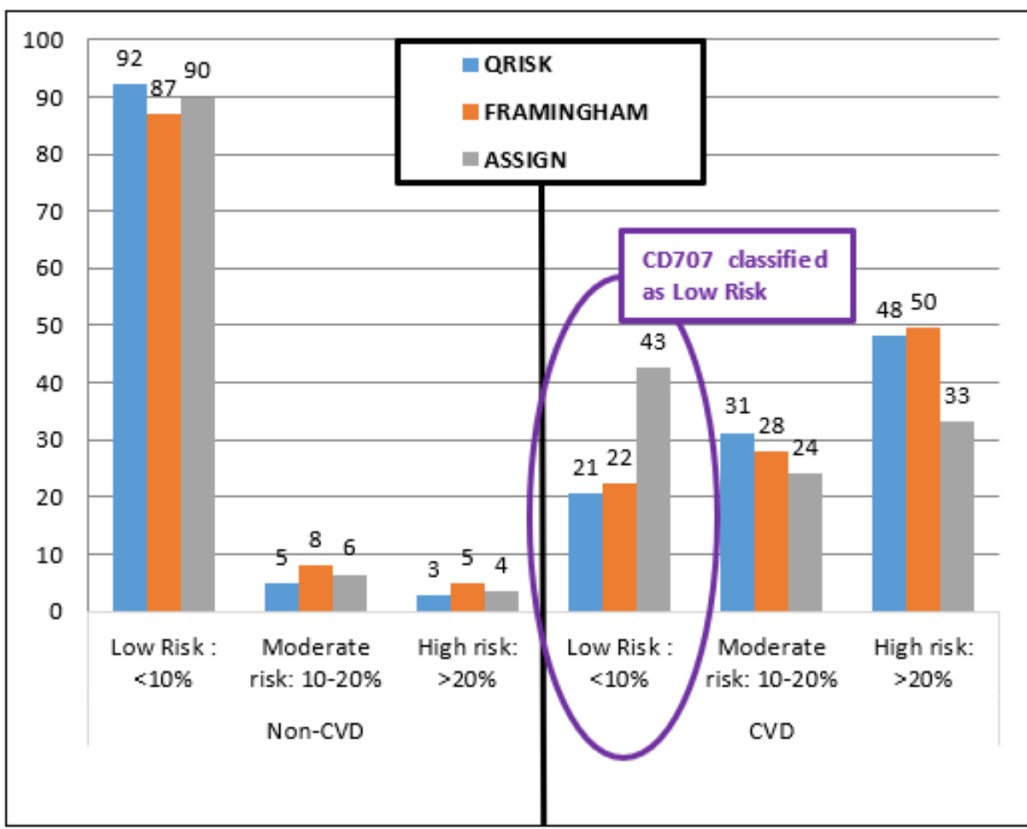

**Figure 6** Percentage distribution of persons categorized by three established risk models into different risk levels for non-CVD and CVD groups.

and evaluation of population-specific tools to enhance CVD management and health outcomes in multi-ethnic populations.

Future work should focus on continuous data collection from this population and the refinement of the model to improve classification accuracy. Additionally, expanding the scope of research to include larger and more diverse datasets could further validate and enhance the robustness of the LinkRisk and DiscrimRisk models. Investigating the integration of advanced machine learning techniques and real-time data analytics could also contribute to the development of more dynamic and adaptive models. These efforts will be instrumental in advancing personalised CVD management strategies and improving health outcomes across diverse populations.

## Funding

This work was supported by The University of Trinidad and Tobago, Postgraduate Department. The funders had no role in study design, data collection and analysis, decision to publish, or preparation of the manuscript.

## Grant Disclosures

The following grant information was disclosed by the authors:
The University of Trinidad and Tobago, Postgraduate Department.

## Competing Interests

Valerie A. Stoute owns V.A.S Consulting.

## Author Contributions

- Amalia Hosein conceived and designed the experiments, performed the experiments, analyzed the data, prepared figures and/or tables, authored or reviewed drafts of the article, and approved the final draft.
- Valerie Stoute conceived and designed the experiments, analyzed the data, authored or reviewed drafts of the article, and approved the final draft.
- Natasha Singh conceived and designed the experiments, prepared figures and/or tables, authored or reviewed drafts of the article, and approved the final draft.

## Human Ethics

The following information was supplied relating to ethical approvals (i.e., approving body and any reference numbers):

University of Trinidad and Tobago and the Ministry of Health of Trinidad and Tobago granted Ethical approval to carry out this study within its facilities (Ethical Ref. No. UTTO/68/13).

## Ethics

The following information was supplied relating to ethical approvals (i.e., approving body and any reference numbers):

University of Trinidad and Tobago Institutional Review Board, Trinidad and Tobago Ministry of Health Ethics Committee (UTTO/68/13).

## Data Availability

The raw data are available at Figshare: Hosein, Amalia (2024). Data file for CVD new models paper TT2015 casecontrol.csv. figshare. Dataset. https://doi.org/10.6084/m9.figshare.21551550.v1.

## Supplemental Information

Supplemental information for this article can be found online at http://dx.doi.org/10.7717/peerj.17948#supplemental-information.

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
