# Peer review of "A classification system for identifying persons with an unknown cardiovascular disease (CVD) status for a multiracial/ ethnic Caribbean population"

_PeerJ, doi:10.7717/peerj.17948_

## Round 0.1 · original submission · Major Revisions

Dear authors,

A decision has been made regarding the following manuscript: A classification system for identifying persons with an unknown cardiovascular disease (CVD) status for a multiracial/ethnic Caribbean population,

Major revisions are necessary to improve your manuscript and, bring it closer to being published. So, please, follow the reviewers' comments and submit the document again.

Regards,

Reviewer 1 ·

Basic reporting

Evaluating Cardiovascular Disease (CVD) risk scores for participants with known CVD and non-CVD in a multiracial/ethnic Caribbean sample
Thank you for the opportunity to review this manuscript titled “Evaluating Cardiovascular Disease (CVD) risk scores for participants with known CVD and non-CVD in a multiracial/ethnic Caribbean sample” which need for clasification systems for Cardiovascular Disease (CVD) that is population specific is important towards understanding the clinical disease and diagnostics associated with the disease. This paper presents the form and validation results of this classification system. I welcome studies that introduce novelty and applicability the importance of introduce new software’s and perspectives of study. I am open to be persuaded to deep understand this relationship.

Experimental design

No comment

Validity of the findings

At face value, it appears that the study design was well-thought out and accurately replicates what could conceivably be implemented in practice. In addition, this study provides consistency with these new technologies and provide knowledge information about his use. This study is particularly important because the publications on this topic are increasingly, so it is necessary to group all these papers to have a better understanding of the results obtained previously. For this reason, it has a valuable and socially useful content. The manuscript is correctly written and structured. I have some suggestions for the authors.

Additional comments

1. For future investigations should be indicated at the end of the abstract as well as in the conclusions section some directions about the results in relation to your findings.

2. The most recent updates should be mentioned in the Introduction? check: it would be interesting if the bibliography was updated.

3. The introduction doesn’t explain the basic concepts used in the paper. Some of these are presented in the methodology section, in terms of how they were measured, however it would have been necessary to clarify their general content in the introduction section. Trully, I think that you can use information of this paper “Hosein A, Stoute V, Chadee S, Singh NR. Evaluating Cardiovascular Disease (CVD) risk scores for participants with known CVD and non-CVD in a multiracial/ethnic Caribbean sample. PeerJ. 2020;8:e8232. Published 2020 Mar 9. doi:10.7717/peerj.8232” to avoid overlap information in both papers. So, will be interesting said why this study apport new information.
The methodology is not clearly explained and justified.
- More information is needs about the group (Inclusion and exclusion criteria)
- Can you more information about data collection to elucidate the procedure?
- Maybe, you can add a schematic representation to elucidate the procedure.
- I personally think that see the invidual values of each participants will be interesting to obtain references values. For this reason, add information tables with individual values, confidence interval 95% Upper, lower. For example, in table I. In addition, add under the table level of significance p .001** or p.01*
- Please improve table 3 and simplify the summary.
- Table 4,5, add confidence interval 95% Upper, lower. And info under the table.
- Can you improve the quality of Figure 1?

From my point of view, the main strength of the manuscript is the results obtained. The analysis revealed a clear and the values are very strong.

Discussions is appropriate, making reference to the results of other studies, but the theoretical and practical implications of the research are vaguely mentioned. It necessary improve it because is a key factor of this work.

The conclusions respond to the objectives but prospective should be included. In addition, the conclusion is very long, can you pass information of your conclusion to the discussion section?

Annotated reviews are not available for download in order to protect the identity of reviewers who chose to remain anonymous.

·

Basic reporting

The article should include a well-defined introduction and background Relevant prior literature is not much included .

Experimental design

The method is not clearly mentioned.
Binomial and logistic regression is existing approaches. What is the contribution of the authors?
The conclusion part is not well defined. what is the outcome of the paper? how this can be extended further?
The introduction part should be well explained. The explanation should be from a broad perspective to narrow down the concepts.
There is no organization of the paper. Which section describes which concepts?
The paper should be systematically organized and the concepts should be explained well.
There is no algorithm or step-by-step procedure to explain the method.
How these classification methods have been applied to the data set?
No flow diagram which shows step by step.
How the results have been measured?
No confusion matrix or formulas for the measures.
Need explanation to results. Only table data is not sufficient.
The conclusion should be explained with data.

Validity of the findings

The conclusions should be appropriately stated and should be supported by the results

Additional comments

there is no algorithm or flow diagram which explains the method properly.

Reviewer 3 ·

Basic reporting

I would also recommend the addition of a flow chart to illustrate the number of patients included and excluded in the study. This would provide a clear visual representation of the study sample and improve the transparency of the study design.

Line 423 - 425 about roc curve is redundent.

Please report ROC with 95% CI, not smallest CI.

Experimental design

The validation results of the classification system could be more robust if additional external validation studies were performed. The results of such studies would increase the generalizability of the findings and provide a better understanding of the accuracy of the classification system. Might be a future study.

Validity of the findings

The alignment of the training, test, and overall accuracy in Table 2 for both Model 1 and Model 2. I recommend the authors clarify and improve the presentation of these results for better understanding and interpretation.

---

## Round 0.2 · accepted · Accept

Dear Dr. Hosein and Dr. Stoute,

I am writing to inform you that your manuscript - A classification system for identifying persons with an unknown cardiovascular disease (CVD) status for a multiracial/ethnic Caribbean population - has been Accepted for publication. Congratulations!

Reviewer 1 ·

Basic reporting

Thank you very much.

The work done exponentially improves the work initially presented. Therefore I recommend its acceptance.

Experimental design

The study's experimental design appears to be well-structured and methodologically sound

Validity of the findings

In summary, the validity of the findings in this study is supported by strong construct and internal validity, appropriate statistical analysis, and practical implications for athletic training. In fact, regarding sample size and generalizability highlight the need for further research to confirm and expand upon these results

Annotated reviews are not available for download in order to protect the identity of reviewers who chose to remain anonymous.

Reviewer 3 ·

Basic reporting

The authors have addressed all my concerns.

Experimental design

The authors have addressed all my concerns.

Validity of the findings

The authors have addressed all my concerns.